# Adaptive Beamforming Approaches to Improve Passive Radar Performance in Sea and Wind Farms’ Clutter

**DOI:** 10.3390/s22186865

**Published:** 2022-09-10

**Authors:** Javier Rosado-Sanz, María Pilar Jarabo-Amores, David De la Mata-Moya, Nerea Rey-Maestre

**Affiliations:** Signal Theory and Communications Department, University of Alcalá, 28801 Alcalá de Henares, Spain

**Keywords:** passive radar, beamforming, interference suppression, direction of arrival

## Abstract

This article presents the problem of passive radar vessel detection in a real coastal scenario in the presence of sea and wind farms’ clutter, which are characterised by high spatial and time variability due to the influence of weather conditions. Deterministic and adaptive beamforming techniques are proposed and evaluated using real data. Key points such as interference localisation and characterisation are tackled in the passive bistatic scenario with omnidirectional illuminators that critically increase the area of potential clutter sources to areas far from the surveillance area. Adaptive beamforming approaches provide significant Signal-to-Interference improvements and important radar coverage improvements. In the presented case study, an aerial target is detected 28 km far from the passive radar receiver, fulfilling highly demanding performance requirements.

## 1. Introduction

A Passive Radar (PR) is a system that has the ability to detect and track targets without transmitting any electromagnetic signal by taking advantage of Illuminators of Opportunity (IoOs) instead of a dedicated one [1]. Some examples are: FM [2], Digital Video Broadcasting—Terrestrial (DVB-T) [3,4,5], Satellite (DVB-S) [6], and Global Navigation Satellite Systems (GNSS) [7,8,9].

In PRs, the lack of electromagnetic emission provides some useful advantages concerning active radars, such as reduced cost, low power supply—which means portability potential—null electromagnetic pollution, and low probability of interception. On the contrary, the exploitation of signals not designed for detection purposes and the null control of transmission parameters make complex signal processing techniques necessary to detect the low energy scattered by the targets.

The use of non-controlled IoOs usually requires PR receiving systems composed of two channels: the reference channel for the acquisition of the direct signal from the IoO, and a surveillance one used to capture the echoes generated by the targets in the surveillance area when they are illuminated by the IoO.

Signal processing is based on the correlation of Doppler-shifted copies of the reference signal, $s_{ref}\left[n\right]$, with the surveillance signal, $s_{surv}\left[n\right]$, to generate the Cross-Ambiguity Function (CAF) (Equation (Equation 1)). The acquired signal is usually divided into Coherent Processing Intervals (CPIs) of $T_{int}$ s (integration time). The CAF is applied to each CPI, giving as output a Range–Doppler (RD) map which defines the observation space for the detector design [10].
(1)$$S_{CAF}\left[m,p\right]=\sum_{n=0}^{N-1}s_{ref}^{*}\left[n-m\right]\cdot s_{surv}\left[n\right]\cdot exp^{-j2\pi \frac{p}{N}n}$$

-*N* is the number of samples ($N=T_{int}\cdot f_{s}$; $f_{s}$ is the sampling frequency).-*m* represents the time bin associated with a delay $m=f_{s}$.-*p* is the Doppler shift corresponding to $f_{dop}=f_{s}\left(p=N\right)$.

Figure 1 presents the CAF generation for a passive radar with a single radiating element surveillance antenna.

Figure 2 shows a general view of the radar scenario. The IoO illuminates the area of interest (sea) and all the potential clutter sources (urban areas, motorways, train tracks, wind farms, harbour, airports, etc.). Although main ground clutter echoes are located at the back of the surveillance antenna, from bearings of very low sensitivity, their intensities are significantly higher than those generated by the vessels to be sought, and will correlate with the reference signal, generating strong contributions compared to target ones.

The bistatic range is defined as the difference between the paths IoO-target-PR and the direct path IoO-PR: $R_{B}=R_{IoO-target}+R_{target-PR}-R_{IoO-PR}$. The bistatic velocity is defined in terms of Doppler shift and operation frequency: $v_{B}=f_{D}\cdot \lambda$. The resolution of the system is defined by the acquisition bandwidth and the integration time for CAF processing: the bistatic range resolution depends on the acquisition bandwidth ($\Delta R_{B}=c/BW_{acq}$), while Doppler resolution depends on the integration time ($\Delta f_{D}=1/T_{int}$). The bistatic velocity is calculated as the addition of the target velocity components along the target-IoO and target-PR directions, denoted as $v_{IoO}$ and $v_{PR}$, respectively, which is $v_{B}=v_{IoO}+v_{PR}$.

One of the main PR interferences, besides clutter, is the Direct Path Interference (DPI) or direct IoO signal acquired by the surveillance channel. It correlates with the reference signal and produces a high level peak at the CAF origin, and raises the pedestal throughout the bistatic range-Doppler domain. On the other hand, CAF contributions of echoes generated by non-desired stationary targets (surface clutter, for example) present maximum values along the zero Doppler line and increase the pedestal throughout the CAF domain. Different techniques for DPI and zero Doppler interferences cancellation have been studied in the literature, such as Extensive CancellationAlgorithm (ECA) [11] and block Normalised Mean Least Squares (NMLS) [11,12,13,14]. DPI can also be reduced through the antenna system by designing an antenna with low gain in the direction where IoO is located, or introducing a null via beamforming techniques if an array is being employed [14].

In most applications, radar operation covers detection and localisation tasks. The CAF maps targets and interference contributions in the range Doppler plane. Bearings can be estimated from Direction of Arrival Techniques (DoA) based on the use of surveillance array antennas in bistatic configurations and/or multi-static configurations involving a set of IoOs and/or a set of PRs. The DoA can be estimated using monopulse techniques or by employing arrays of antennas. The advantages of employing digital phased arrays in combination with array processing techniques are numerous, from beam shape flexibility and simultaneous beam generation to jamming and clutter suppression without forgetting the angle estimation accuracy [15]. Multiple beams can be generated and a CAF can be calculated for each beam, but to exploit the mapping generated by the CAF, a CAF can be generated for each single radiating element (using the acquired signal and the reference one), and beamforming coefficients can be calculated in the transformed CAF domain for different (range, Doppler) pairs.

Adaptive Beamforming (ABF) is commonly employed in active radars to suppress jamming signals enabling target detection [16]. Deterministic and adaptive beamformers have also been designed to improve target detection and localisation performances [4,5,17,18]. ABF techniques are examples of optimisation problems that can take advantage of methods proposed for a wide variety of applications [19,20,21,22].

In the presented paper, ABF techniques are applied for clutter characterisation and rejection in complex PR scenarios. IDEPAR (Improved DEtection techniques for PAssive Radars) is the PR demonstrator developed at the University of Alcalá [3,4,5]. It was deployed in a coastal scenario located in the South of Spain, in an area characterised by intense ground and sea clutter returns. Ground returns are mainly generated by urban areas, motorways with high amounts of traffic, train tracks, a harbour, a military base, and wind farms. Sea states and wind turbines’ configuration highly depend on weather conditions, giving rise to complex and time-space varying clutter contributions. Due to the use of IoOs, for which the illuminated areas are defined by the radiocommunication service they belong to, clutter sources extend far from the PR surveillance area, requiring a specific methodology that considers strong clutter sources out of the PR surveillance area, that are illuminated by the IoO and can receive a more intense signal than the vessels to be sought. This is the case of the wind farms located between the IoO and the PR, at the back of the PR surveillance antenna. Two solutions have been proposed: one based on finding the DoA of those sources and generating nulls in the array pattern, and another based on adaptive techniques. In the considered case study, usual Front-to-Back surveillance antenna requirements can be unsuitable due to the high bistatic radar cross section of the clutter sources, and the power radiated by the omnidirectional IoO. Regarding the second solution, two approaches are studied to estimate the interference covariance matrix, avoiding the inclusion of targets in the training or sample data, which is the key point for achieving good performance in adaptive beamforming.

The rest of the paper is structured as follows: the complex coastal scenario and the IDEPAR demonstrator are described in Section 2. Section 3 proposes two approaches to reject strong interferences applying beamforming techniques: through deterministic nulling and employing adaptive techniques. Methodologies to apply these techniques has been described and solutions to estimate interference characteristics have been proposed. Results prove that both approaches allow for to rejection of most interferences in real data, and adaptive beamforming considerably improves the targets’ SNR, which will be translated into an improvement in the detection capability. Section 3 also estimates the direction of arrival of targets under study and presents the results overlapped in the real scenario to validate those targets’ tracks. Section 4 summarises the main contributions and results achieved in this article.

## 2. Problem Formulation

Currently, IDEPAR consists of eight acquisition channels with a maximum bandwidth of 100 MHz and operating frequencies ranging from 40 MHz to 6 GHz. Within the possible opportunity signals available in this range, DVB-T has been selected for the research carried out in this article.

One channel is dedicated to reference signal acquisition, and the other seven are used to acquire the signals captured by the seven single radiating elements of the surveillance array. The reference channel employs a commercial antenna *Televes DAT 75* which provides high directivity. The surveillance antenna is the Non-Uniform Linear Array (NULA) described in [5,23] and shown in Figure 3. The beam-width of these antennas at DVB-T frequencies is $\pm 42^{\circ }$ in the azimuth plane, so the total field of view of the system employing the array is $84^{\circ }$. The main parameters of the array were obtained for two steering angles: $Phi_{sa})=0^{\circ }$ and $Phi_{sa})=\pm 30^{\circ }$. Table 1 summarises the array gain ($G_{\Phi_{sa}=0^{\circ }}$ and $G_{\Phi_{sa}=\pm 30^{\circ }}$), the Half Power BeamWidth, HPBW, ($BW_{3dB\_\Phi_{sa}=0^{\circ }}$ and $BW_{3dB\_\Phi_{sa}=\pm 30^{\circ }}$), and the SideLobe Level ($SLL_{\Phi_{sa}=0^{\circ }}$ and $SLL_{\Phi_{sa}=\pm 30^{\circ }}$).

Array signal processing techniques are applied in the CAF transformed domain [4]. The reference signal is used to generate seven CAFs, one for each surveillance single radiating element. Figure 4 depicts the processing architecture. The CAF generation is included in the Processing Stage. For each [m,p] pair, a snap shot is obtained that will be applied to DoA and beamforming techniques. Digital beamforming is applied, applying the corresponding set of weights w to each to each snapshot to form a beam in the desired steering direction, fulfilling maximum directivity. Next, five-orthogonal beams within the single element HPBW are generated. Those beams are steered to azimuth angles $\phi_{sa}=\left[-30.5^{\circ },-14.5^{\circ },0^{\circ },14.5^{\circ },30.5^{\circ }\right]$. Due to the sign convention applied, negative steering beams are at the right of bore-sight in the surveillance area seeking direction. The output of each beam is obtained following Equation (Equation 2), where X[m,p] is a matrix of *N*-elements rows times the number of total samples of each RD map (RD cells) (Equation (Equation 3), where the subscript indicates the element or antenna that is used in each CAF) and Y[m,p] is the output RD map for each desired beam.
(2)$$Y\left[m,p\right]=w^{H}X\left[m,p\right]$$
(3)$$X=\left[s_{CAF}^{1}\left[m,p\right];s_{CAF}^{2}\left[m,p\right];\cdots ;s_{CAF}^{N}\left[m,p\right]\right]$$

The measurements considered in this article were carried out during trials in a coastal scenario in the south of Spain (Cádiz). The main elements of the scenario that can be seen in Figure 5 are:The Area of Interest (AoI) or surveillance area defined by the surveillance antenna broadside direction ($270^{\circ }$N) and the single radiating element HPBW. It covers the main naval routes from and to Cádiz harbour and includes the approaching route of aerial targets to Rota Military Base airport, with a huge number of communication and radar systems.The IoO is located next to Jérez de la Frontera, at a distance of 21 km from PR location. The central acquisition frequency is tuned to 774 MHz, comprising two DVB-T channels of around 8 MHz each.Wind farms areas which are harmful interference sources due to the high Radar Cross Section (RCS) of wind turbines, up to 60 dBsm [24,25,26].A set of roads just at the back of the PR system, with ground targets that are considered to interfere, as they are not expected in the defined area of interest.

The radiation pattern of the two beams under study in this article is also superimposed in the scenario: beam 1 steered to −30.5${}^{\circ }$ and beam 3 steered to 0${}^{\circ }$. The side lobes due to array configuration can be appreciated. Side lobes, even providing considerably less gain than the main lobe, can cause the detection of targets in more than one beam, i.e., not only in the beam whose HPBW contains the target’s direction. Side lobes are also responsible for interferences coming from different directions than the main one; for example the wind turbines in the considered scenario, which worsen the detection capability.

Taking into consideration targets’ dynamics, CPIs of 500 ms were generated with a Pulse Repetition Interval (PRI) of 250 ms, and processed for Zero-Doppler interference cancellation using an ECA filter, and CAF generation. Seven CAFs (one for every single radiating element of the surveillance array) were obtained, and the set of orthogonal beams was generated in this transformed RD domain. Figure 6 shows the maxima of the CAFs for the rightmost beam ($-30.5^{\circ }$ with respect to the array boresight, $300.5^{\circ }$N) and the central one (at the boresight, $270^{\circ }$N). These two beams under study are represented in Figure 5 overlayed in the scenario. Maxima were calculated along a block of 160 consecutive CPIs. The first beam contains the area closest to the harbour where more targets are present. There are a lot of high-power scatterers that group during the acquisition time, forming trajectories in this beam, whereas no targets are appreciated in beam 3.

The main problems observed in these RD maps are the high-level lines located at different ranges that extend along the Doppler dimension crossing the zero-Doppler. These structures not only can mask low-Doppler targets at the same ranges, but can also increase the CAF pedestal level, reducing the SIR of targets far from the position of the lines in the CAF. Figure 7 shows a more detailed analysis of beam 3. This beam is interesting due to the absence of targets of interest. In Figure 7a, four main clutter contributions are observed. One is around range bin 156. The second one, which is more dispersed, extends from range bin 296 to bin 353; another one appears at range bin 605, and the last one appears at range bin 935. Figure 7b depicts isorange contours for these range bins and allows the identification of wind farms as potential interference sources. This cannot prove that those interferences are due to wind farms and not due to sea clutter, as they can not be decoupled, but the coincidence of iso-range curves with the areas of wind turbines indicates that these may be the sources.

In addition to the increase in the RD map background and possible targets undercover, another problem is the non-stationary character of interference sources over time. In Figure 8, three different CPIs of beam 1 are presented to show the time variation of interference contributions in range and Doppler spreading. These time variations complicate the design of clutter maps or other clutter filtering techniques. Beamforming techniques can reject interference sources in the spatial domain. Decreasing the power of interferers will also decrease the background level of the RD map, increasing detection performance.

## 3. Interference Suppression through Beamforming Techniques

Beamforming techniques can be useful to spatially filter interference sources, improving the performance of a PR system. In bistatic scenarios such as the one described in Section 2, strong interference sources can be distributed out of the area of interest, and can generate radar returns stronger than those produced by desired targets. Their characteristics can vary along time, but with effects that persist during system operation. This situation is the result of using non-cooperative IoOs, which are not designed for the surveillance application of interest. Because of this, the generation of deterministic nulls along bearings at the back of the surveillance antenna array and the possibility of adapting these nulls to variations in interference properties are studied [27,28].

### 3.1. Deterministic Null Generation

The first proposal to limit the likelihood of the performance worsening due to interferences consists of estimating the DoA of main sources and generating deterministic nulls in the estimated angles.

The first step is to select the RD cells of a maximum return due to interferences. For this purpose, as an ECA filtering was applied in the first processing stage, the first non-zero Doppler row is selected. In this case, we start from positive Doppler values and, taking into account the selected integration time of 500 ms, the first selected value is 2 Hz. According to beamforming theory, N-1 nulls can be generated using an array of N elements citeVanTrees. Taking this into consideration, the N-1 greater peaks of the CAFs intensities along the range dimension for the selected Doppler of 2 Hz are determined. The value of the RD map along all the ranges for the selected Doppler are plotted and the peaks are selected for each antenna. The selected peaks of N elements are combined and the N-1 peaks of maximum level are selected, ensuring that they have different interference sources, i.e., they are reasonably spaced in range.

The DoA estimation can be achieved in different ways, each of them with its advantages and drawbacks. An estimation through monopulse techniques can be performed via amplitude comparison, forming sum and difference beams from the array elements, or through phase comparison (interferometry). These techniques are fast and involve low computational cost, but they can only estimate the DoA within the half-power beam-width, so they are not appropriate, bearing in mind the scenario knowledge that suggests that interferences can come from the back part of the array if they are caused by the wind farms. The selected technique for estimating DoA is through beam scanning in all possible directions of the azimuth plane following a discrete grid in steps of $1^{\circ }$: a beam steered to each direction is generated, and the power received in the evaluated RD cell is calculated for each beam and the direction associated with the beam generating the peak of power is assigned as the angle of that interference source. This technique for estimating DoA involves a high computational load, as each RD cell must be evaluated in all the set of beams by applying the corresponding weights in each case. One problem with this technique is that generating beams steered to directions outside the visible margin of the array causes the appearance of grating lobes that involve ambiguity in DoA estimation.

Once the angles of the interferences are known, the optimum weights $w_{o}$ for generating a zero-order null in the desired direction $\phi_{n}$, for n=1,⋯,(N−1), are calculated following Equation (Equation 4), with $v_{k}\left(\varphi_{n}\right)$ being the steering vector of the array for the null constraints $C=[v_{k}\left(\phi_{1}\right)\vdots v_{k}\left(\phi_{2}\right)\cdots v_{k}\left(\phi_{N}\right)]$ and $w_{d}$ the weights for achieving a desired pattern; for example beam steering, prior to null generation [28].
(4)$$w_{o}^{H}=w_{d}^{H}\left(I_{N}-C{\left[C^{H}C\right]}^{-1}C^{H}\right)$$

Figure 9 shows how most of the interference sources found previously in CPI 13 are rejected by applying the deterministic null proposal. However, observing the cumulative maxima of RD for all 158 CPIs after null generation (Figure 10a), it can be determined that the power of the targets has also been reduced, i.e., targets’ trajectories are less appreciable above noise background level. This decrease in targets’ power is caused by the deviation of the direction of the beam with respect to the desired one due to null generation in the vicinity of beam steering, as well as the decrease in directivity due to beam broadening (Figure 10b). The patterns have been represented by employing the simulated pattern of an isolated antenna for all the elements, i.e., coupling effects between antennas in the array formation are not considered.

One of the targets with a clearer trajectory along the processing time (marked with a blue box in Figure 10a) was selected for studying the difference in Signal-to-Noise Ratio (SNR) before and after the application of the proposed technique. The selected target is suspected of being a maritime one, due to their Doppler frequency shift indicating low velocity. SNR has been calculated as the ratio of maximum target power with respect to background level. Figure 11a proves that only in some cases the SNR of a maritime target (the one highlighted with a square in Figure 10a) is greater than it was without the application of this technique.

Sub-optimum detection approaches for approximating the envelope detector are usually applied using a fixed detection threshold. Constant False Alarm Rate (CFAR) techniques are used to maintain the $P_{fa}$ at a constant level despite clutter parameter variations, estimating its unknown parameters from observations acquired by the radar. If some usual detection requirements are imposed, as probability of detection $P_{D}=80\%$ and probability of false alarm $P_{fa}=10^{-5}$, a minimum SNR of 17 dB is required to assure these requirements if a CFAR detector is employed considering Swerling I type for targets under AWGN conditions (Figure 11b) [29]. Required SNR is only achieved in a few CPIs in the case of deterministic nulling and not completely assured in classic beamforming, being the target SNR in most of the CPIs just around this border of 17 dB.

### 3.2. Adaptive Beamforming (ABF)

The second proposal to improve targets’ detectability by decreasing the power of interference sources is adaptive beamforming. From the multiple techniques proposed in the literature for ABF, for this application, Sample Matrix Inversion (SMI) in combination with a Minimum Variance Distortionless (MVDR) optimum beamformer is used [28]. SMI means that the matrix containing the information about the interference for each channel—that is, the covariance matrix—is estimated from sample training data and must be inverted. MVDR is adequate in this case because post-Doppler covariance matrix estimation is performed. MVDR will optimise the weights needed to minimise the power of the sources contained in the sample data. Because of this, a good selection of RD cells for estimation will allow the interferences power to be minimised without cancelling any targets if they are not included in the sample data. The optimum MVDR beamformer weights are calculated from Equation (Equation 5) where $v_{s}$ states for the steering vector and ${\hat{S}}_{n}$ is the estimated interference covariance matrix [30].
(5)$$w_{mvdr,smi}^{H}=\frac{v_{s}^{H}{\hat{S}}_{n}^{-1}}{v_{s}^{H}{\hat{S}}_{n}^{-1}v_{s}}$$

The key point in this technique is the selection of training data from RD maps of all the channels to estimate the covariance matrix. The selected data must contain just interference plus noise. Targets must be excluded from training data to avoid their rejection. Two methods are proposed:The first one takes advantage of the fact that the interference sources expand from zero Doppler and employs the cells surrounding zero Doppler for all ranges, excluding the nearest ones. This is because near ranges contain high-level peaks due to nearby clutter that comes from almost all directions. This proposal has been evaluated by selecting the Doppler rows of ±2 Hz and ±4 Hz. This proposal can include low-velocity targets, or some that, due to geometry, have a low Doppler shift, and therefore cancel it.The second proposal tries to employ all the Doppler information of the interferences, so the range of the main interferences is calculated and the columns corresponding to these ranges, and the neighbouring ones, for all Doppler shifts are selected for covariance matrix estimation. This proposal has the advantage that if a target appears in the same range as one form of interference, it will be only cancelled in one or a few CPIs, as it will change range due to movement.

The SNR loss due to covariance matrix estimation, instead of perfect knowledge of interferences, depends on the number of data samples employed for estimation. In the first proposal, the number of samples employed is equal to $N_{s}=9840$; that is, 4 Doppler cells × (2500−40) range cells. In the second proposal, $N_{s}$ depends on the number of dominant interferences found in each CPI RD map. The total number of samples is $N_{s}=N_{I}\times 500$, with $N_{I}$ interferers or peaks selected (i.e., one range cell is selected for each interferer), and 500 is the number of Doppler cells of an RD map. The number of range cells employed is, in most cases, between 15 and 40 so, in the worst case scenario (minimum value) the number of samples is $N_{s}=7500$. In [28] it is proved that for $N_{s}=1000$, the SINR loss converges to 0 dB, so both proposals have an adequate number of samples needed to achieve a valid covariance matrix estimation.

Both proposals of MVDR ABF achieve a similar level of interference suppression to Deterministic Null (DN) generation (Figure 12): most interference sources are suppressed except those in range bins 350 and 600. In the ABF case, the background level is highly decreased, and non-suppressed interferences seem to be stronger. This phenomenon is even more important in the second proposal for ${\hat{S}}_{n}$ estimation.

Quite the opposite of the DN case; in ABF cases, the targets’ level is not decreased, as it can be checked in the RD map of cumulative maxima (Figure 13). The decrease in background level allows a greater number of targets’ trajectories to be observed. It is interesting to highlight that now, a trajectory appears at ranges further than 2000 bins and around 300 Hz Doppler frequency shift, which is probably due to an aerial target, because that Doppler shift corresponds to velocities too high for maritime vehicles. This trajectory is even clear in the case of the second proposal, i.e., estimation employing all Doppler cells for some specific range gates corresponding to interference ones.

Figure 14 shows the SNR of two targets: the maritime one also shown for DN case in Figure 11a, and the aerial one that appeared after ABF implementation. It is possible to check what was observed in Figure 13: SNR is improved significantly, employing ABF for interference suppression. In the case of the maritime target, the improvement is similar for both proposals, but in the case of the aerial one, the second proposal works better. Now, the SNR values achieved for both maritime and aerial targets in most of the trajectory will assure detection requirements stated if a CFAR detector is employed.

### 3.3. Results Comparison

Two different techniques have been proposed in previous sections to solve the problem of interferences in order to improve targets’ detection. Comparing Figure 10a with Figure 13 and Figure 11a with Figure 14 it can be determined that both solutions reject most of the interferences, but only adaptive techniques improve targets SNR.

Figure 15 serves as a means of comparison or contrast between solutions, as it visually highlights the performance improvement of each of the proposed algorithms. It shows RD maps of cumulative maxima zoomed in target areas for all cases: classic BF, BF with Deterministic Nulls generation, and Adaptive Beamforming for the two proposals of sample matrix estimation. The SNR improvement for both targets is appreciable when adaptive algorithms are employed: the maritime target power is greater and extends to more RD cells, and the aerial one can now be differentiated from the background, and therefore can be detected, as proven through the calculus of SNR and detector requirements.

Table 2 summarises the mean SNR for both targets for each beamforming solution, highlighting the SNR improvement already mentioned in the case of adaptive techniques.

In terms of computation complexity, adaptive techniques are, by nature, more complex than deterministic ones because they are based on the inversion of the covariance matrix. In the considered case study, the surveillance array is composed of seven single radiating elements. The covariance matrix dimension is 7×7 and this inversion is not so costly. On the other hand, deterministic nulls generation has an intrinsically low computational cost, as it only applies a simple equation. However, it has added computational load due to the selection of the N−1 greater peaks due to interferences and due to the direction finding of interferences which involve numerous beam steering through the azimuth plane to calculate interference power in those angles.

### 3.4. Direction of Arrival of Targets under Study

In previous sections, spatial interference suppression techniques were proven to improve targets’ SNR. In this section, the angle estimation capability of the array will be demonstrated by taking advantage of the SNR increase to select the target’s points in each CPI to perform Direction of Arrival (DoA) estimation.

DoA has been performed by means of spatial filtering: generating beams in multiple directions and computing the angle where maximum target scatter is achieved. The estimated angles for the two targets studied in previous sections are shown in Figure 16, as well as the location of those targets in the real scenario during the acquisition time. Observing the estimated angle for the aerial target, it can be determined that, in the first CPIs, the estimation seems to be wrong, as it differs considerably from the angles estimated later on. This is due to the low target SNR achieved in those CPIs, as can be seen in Figure 14. Location of targets confirms the suspects that the farthest target is an aerial one in its approach to the Rota landing strip, but it also allows us to measure the real distance of targets from the PR: the maritime target is around 9 km from the PR, while the aerial one is at distances ranging from 26 km to 28 km from the PR.

## 4. Conclusions

This work presents different solutions based on beamforming techniques, taking advantage of an array architecture designed in previous work to be used for interference suppression in real passive radar data.

The first solution proposes estimating the direction of arrival of the interferences and generating nulls in those directions in a deterministic way. It has been proved that the interferences power was decreased, but the targets’ level also decreased as the beam pattern was eroded. Results showed that this technique does not improve targets’ SNR, but the interference rejection will help to detect targets, close to cells containing interferences, which in another case would be masked.

The second solution proposes applying adaptive beamforming to generate optimum weights employing the MVDR algorithm. Two different techniques were proposed to estimate the interference covariance matrix from RD maps, avoiding the inclusion of targets in training data samples. Both techniques show good performance, suppressing the interferences and improving targets’ SNR significantly. These solutions allowed us to detect a target at a very far range: more than 26 km from the PR, which means around 47 km in bistatic range. This target was not appreciated without ABF implementation, and detection at this range in such clutter and interference conditions is a great achievement.

The problem of beam erosion when generating deterministic nulls may be solved by limiting the proximity of nulls to beam steering or employing some constraints to assure beam steering. The interference suppression performance must be studied if those are employed. However, a more important problem of this proposal regards the implementation in real-time systems: estimation of DoA in all the azimuth plane angles is computationally expensive, and if it must be carried out in each CPI, real-time processing may be compromised. The main problem of ABF solution comes from the need to invert the sample matrix, but this can be solved by employing mathematics techniques such as QR decomposition.

Finally, the improvement in SNR was employed to select the targets and perform DoA estimation, locating them in the AoI of the scenario and validating them as targets of interest.

## Figures and Tables

**Figure 1 sensors-22-06865-f001:**
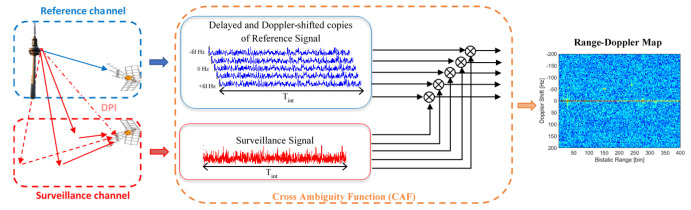
PR geometry scheme and operation principle. Reference signal (blue), surveillance signal components (red): targets’ echoes (continuous), sea clutter (dashed) and Direct Path Interference or DPI (dash and dotted).

**Figure 2 sensors-22-06865-f002:**
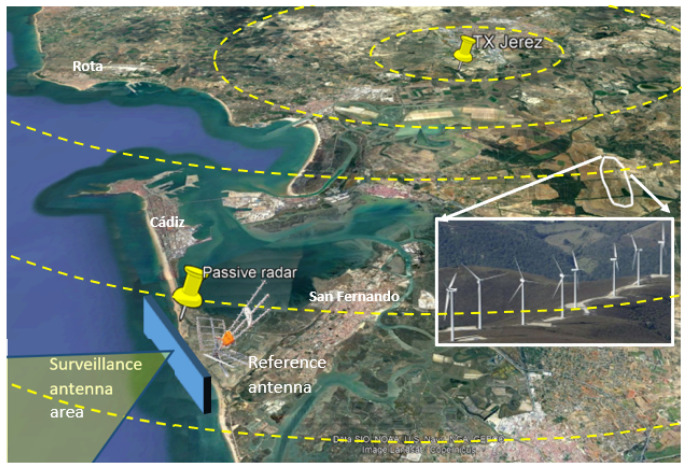
General view of the radar scenario with its main components.

**Figure 3 sensors-22-06865-f003:**
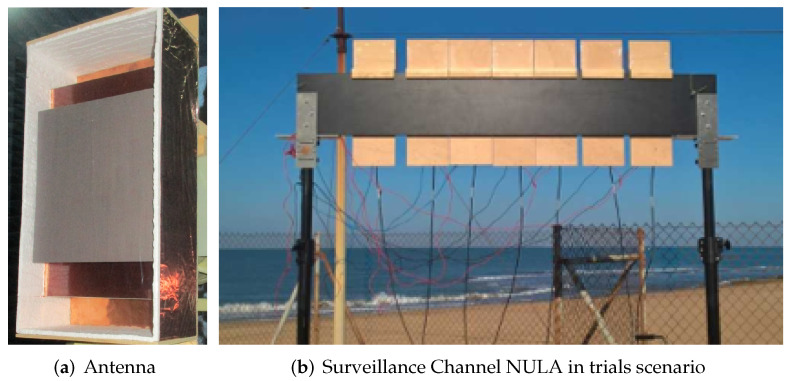
Prototype of single radiating element and back view of Non-Uniform Linear Array employed for the surveillance channel in the trials scenario.

**Figure 4 sensors-22-06865-f004:**
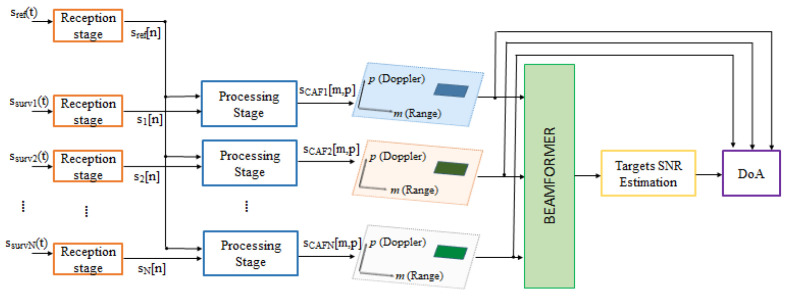
Scheme of processing stages employed showing beamforming processing in range-Doppler domain.

**Figure 5 sensors-22-06865-f005:**
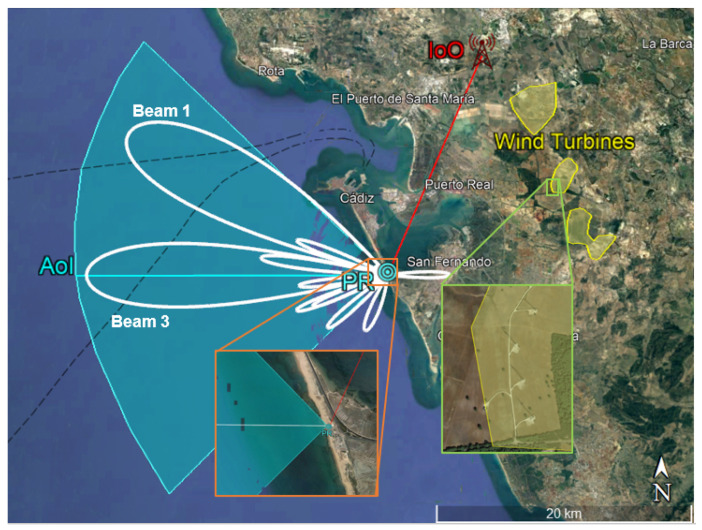
Sketch of the trials scenario located in Cádiz. It shows the IoO and PR locations, the Area of Interest and wind turbines interference areas. The pattern of the two beams under study are also superimposed: beam 1 steered to −30.5${}^{\circ }$ and beam 3 steered to 0${}^{\circ }$.

**Figure 6 sensors-22-06865-f006:**
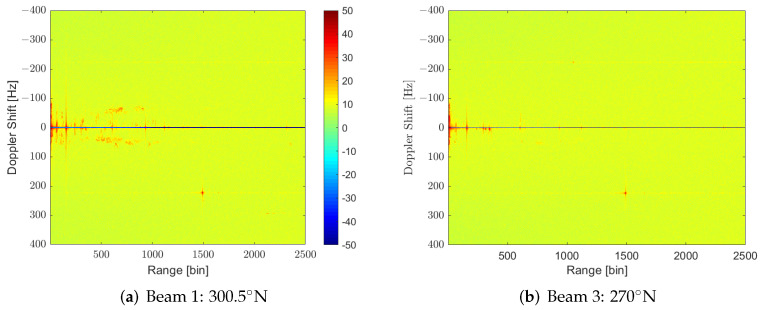
Cumulative of maxima of RD during the acquisition time for two orthogonal steerings.

**Figure 7 sensors-22-06865-f007:**
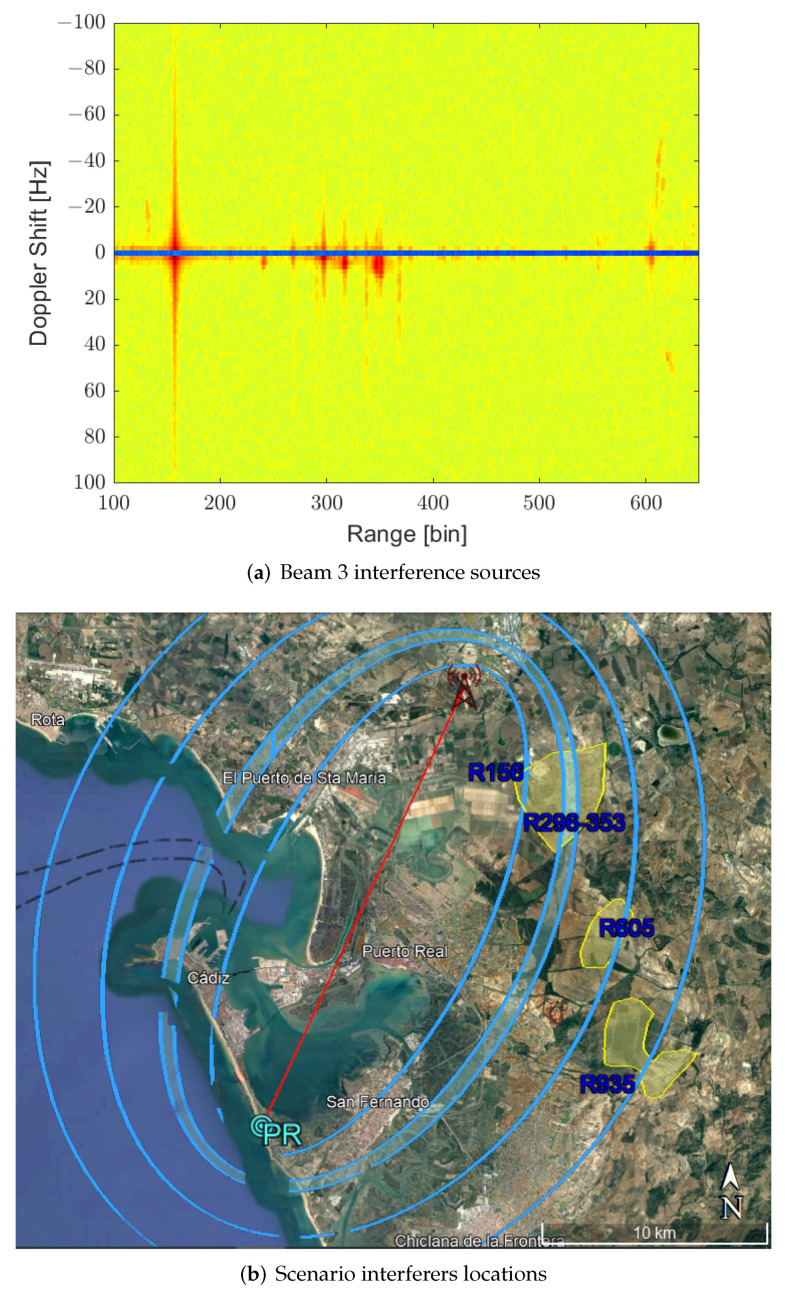
Zoom of cumulative of maxima of RD in beam 3 to appreciate the interference sources and location of these interferers ranges in the scenario.

**Figure 8 sensors-22-06865-f008:**
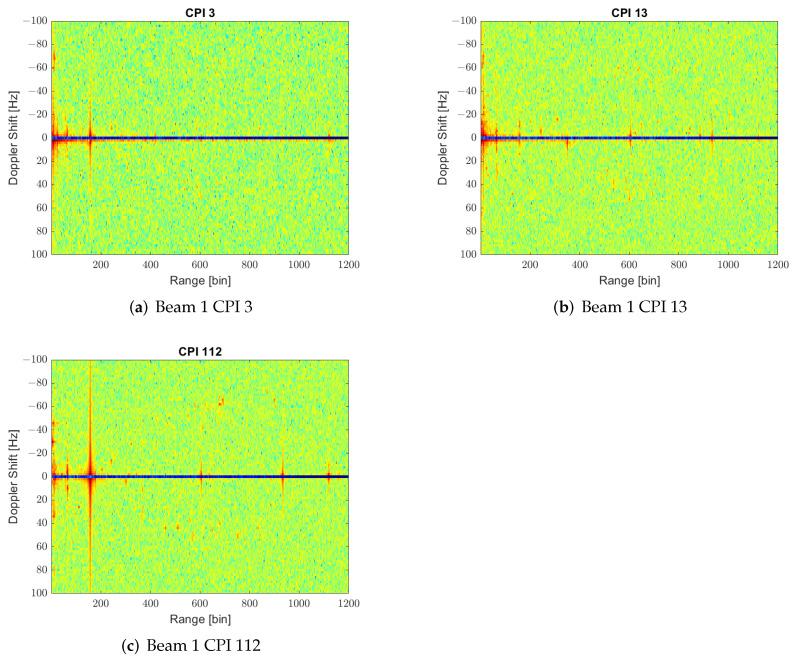
RD maps of different CPIs showing the variance in interference sources through the acquisition time.

**Figure 9 sensors-22-06865-f009:**
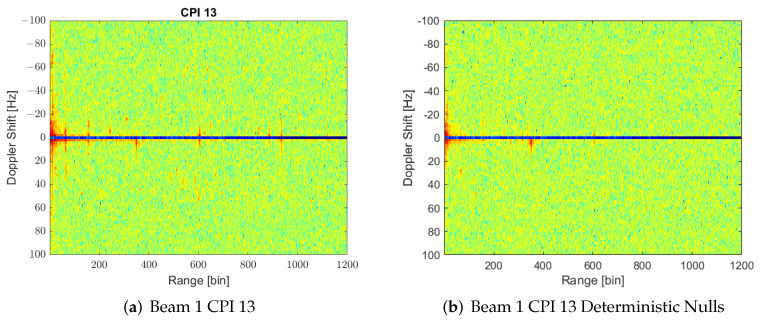
RD maps of CPI 13 showing how most of the interference sources are strongly rejected through deterministic nulling along the estimated DoAs.

**Figure 10 sensors-22-06865-f010:**
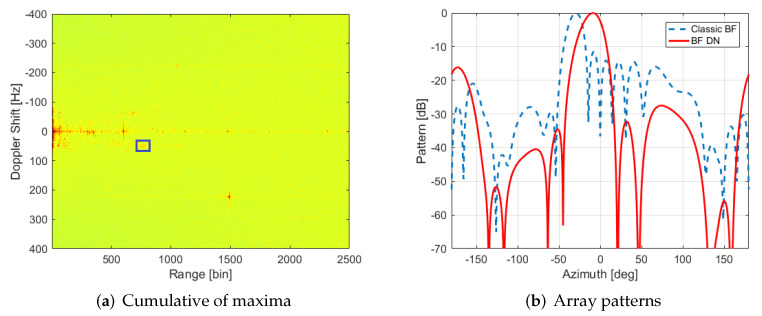
(**a**) Cumulative of maxima through the 158 CPIs in beam 1 with deterministic null generation; (**b**) array pattern for the same beam and CPI 13 comparing the case of classical beamforming with the Deterministic Nulls (DN) generation one.

**Figure 11 sensors-22-06865-f011:**
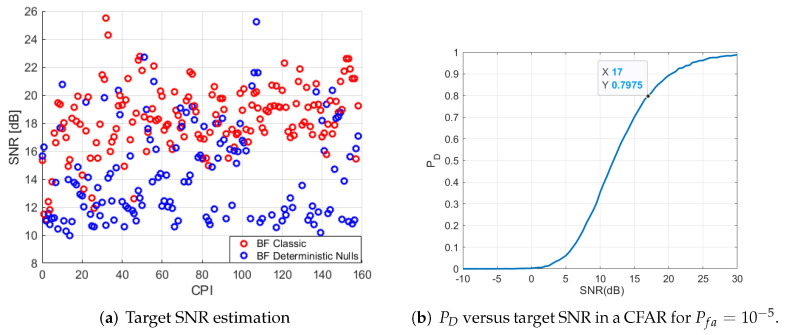
(**a**) SNR of selected targets in each CPI for classical beamforming and deterministic nulls generation; (**b**) curve of required SNR for a given $P_{D}$ and $P_{fa}=10^{-5}$ in case of a CFAR considering Swerling I targets.

**Figure 12 sensors-22-06865-f012:**
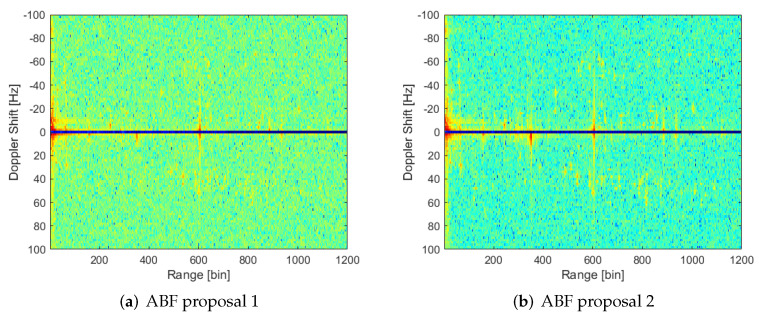
RD map of CPI 13 with adaptive weigths through MVDR algorithm and employing the two proposals for interference covariance matrix estimation.

**Figure 13 sensors-22-06865-f013:**
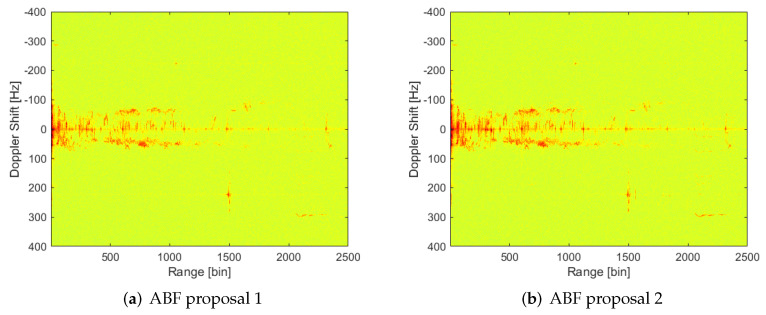
Results of processing the 158 CPI with adaptive weights through MVDR algorithm generating first beam and employing the two proposals for interference covariance matrix estimation.

**Figure 14 sensors-22-06865-f014:**
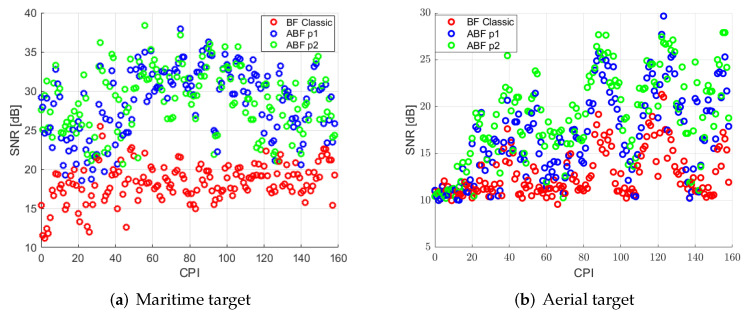
SNR of selected targets in each CPI for classical beamforming and ABF proposals.

**Figure 15 sensors-22-06865-f015:**
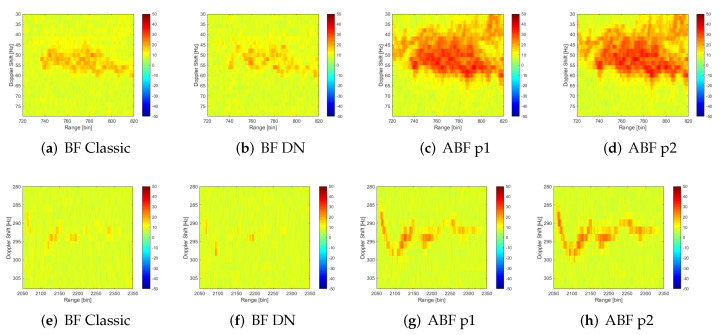
Cumulative of RD maxima zoomed at targets regions (**a**–**d** maritime target and **e**–**h** aerial one) for classic beamforming and the solutions proposed for interference suppression: deterministic nulls generation (DN), adaptive beamforming with sample matrix estimation following first proposal (ABF 1) and second proposal (ABF 2).

**Figure 16 sensors-22-06865-f016:**
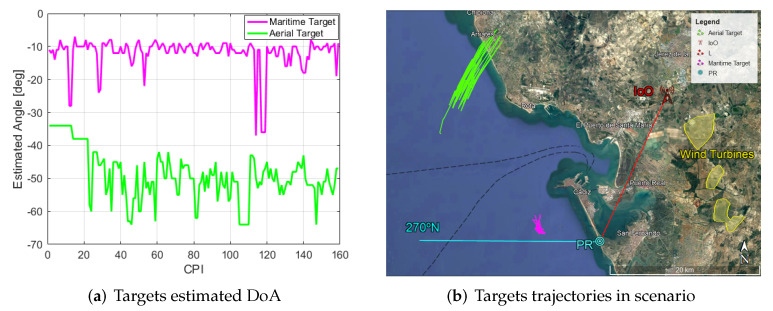
Estimated angle of each target along CPI and transformation of characteristics extracted from the radar data (range, angle) from them to Google Earth in the trials scenario.

**Table 1 sensors-22-06865-t001:** Main parameters of the surveillance array (Figure 3) [5]. $G_{\Phi_{sa}=0^{\circ }}$ and $G_{\Phi_{sa}=\pm 30^{\circ }}$ are the array gain at steering angles (sa) $0^{\circ }$ and $\pm 30^{\circ }$, respectively, $BW_{3dB\_\Phi_{sa}=sa^{\circ }}$ is the half power beamwidth (HPBW) for both steerings, and $SLL_{\Phi_{sa}=sa^{\circ }}$ is the SideLobe Level.

$G_{\Phi_{sa}=0^{\circ }}$	15.2 dBi
$BW_{3dB\_\Phi_{sa}=0^{\circ }}$	$13.6^{\circ }$
$SLL_{\Phi_{sa}=0^{\circ }}$	14.9 dB
$G_{\Phi_{sa}=\pm 30^{\circ }}$	14.4 dBi
$BW_{3dB\_\Phi_{sa}=\pm 30^{\circ }}$	$15.5^{\circ }$
$SLL_{\Phi_{sa}=\pm 30^{\circ }}$	13.6 dB

**Table 2 sensors-22-06865-t002:** Mean SNR of targets for each proposed solutions along all the acquisition.

	Maritime Target	Aerial Target
BF Classic	18.61 dB	13.16 dB
BF DN	15.01 dB	11.95 dB
ABF p1	29.69 dB	18.15 dB
ABF p2	29.73 dB	20.14 dB

## Data Availability

Not applicable.

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
