# Peer review of "Adaptive Beamforming Approaches to Improve Passive Radar Performance in Sea and Wind Farms’ Clutter"

_sensors, 2022, doi:10.3390/s22186865_

Round 1
Reviewer 1 Report
Authors presented a novel adaptive beamforming solution to mitigate interference to a passive radar due to sea and wind-farm. The result is interesting. However, a few points should be addressed carefully to strengthen the work.
1. “Currently, IDEPAR consists of 8 acquisition channels with a maximum bandwidth of 100 MHz and operating frequencies ranging from 40 MHz to 6 GHz.”
In figure 2(b), there are only 7 channels installed on the system. Components of the system should be labeled clearly in the figure.
2. In Problem formulation section, authors should describe detailed method instead of just referring to the reference [5].
3. In figure 4, only beam 1 and beam 3 are displayed. Is beam 1 referring to azimuth angle of 0 and beam 3 referring to azimuth angle of -30.5 degree?The figure should be labeled clearer.
4. In figure 4, do small lobes refer to side lobes due to array configuration? If so, how does side lobe affect detection?
5. Is interference due to sea or wind-farm clutter. Can the factors be decoupled?
6. In figure 4, the Radar is facing the sea, and wind turbines are not in the detection region. In this layout, what is mechanism that wind turbines interfere doppler range?
7. It is unclear how were experiments of detecting maritime target and aerial target conducted.
8. Comparison of the Range-Doppler maps of targets with and without proposed algorithms is absent. This important information should be included to straightforward demonstrate performance of the proposed solution.
9. in Figure 14, maritime target and aerial target is hard to differentiate due to very similar color (bright yellow and bright green).
Author Response
Thank you very much for your comments and suggestions. Point-by-point answers are provided in the attached document and included in the new version of the article.

Reviewer 2 Report
The results look encouraging and motivating. But there are still some contents, which need be revised in order to meet the requirements of publish. A number of concerns listed as follows:
(1) The abstract should be narrow down on the problem and highlight the need of the proposed work with experimental results
(2) In the introduction section, you should give the novelty and the contributions of your works.
(3) The method/approach in the context of the proposed work should be written in detail.
(4) How about the computation complexity of the proposed method?
(5) At Line 54, “In the present paper,…”—> In the presented paper,…”…. There are some grammatical mistakes and typo errors. Please proof read from native speaker.
(6) The framework of the proposed method should be provided.
(7) The literature review is problematic. The authors discussed the importance. In contrast, the review on technical development is insufficient. For example, 10.1109/JSTARS.2021.3059451 ; 10.3390/agriculture12060793; 10.1007/s10489-022-03719-6 ï¼›10.1016/j.engappai.2022.105139 and so on.
(8) The effectiveness of the proposed method needs to be verified by contrast experiments.
(9) The authors need to interpret the meanings of the variables
(10) Figures quality need to be enhanced to better understanding.
Author Response

(The authors gave the same response as above.)

Round 2
Reviewer 1 Report
The authors have addressed all questions well. I recommend publishing the revised manuscript.
Reviewer 2 Report
I have appreciated the deep revision of the contents and the present form of this manuscript. All my previous concerns have been accurately addressed. I think that this paper can be accepted.